# SyncTalklip: Highly Synchronized Lip-Readable Speaker Generation with Multi-Task Learning

## ABSTRACT

Talking Face Generation (TFG) reconstructs facial motions concerning lips given speech input, which aims to generate high-quality, synchronized, and lip-readable videos. Previous efforts have achieved success in generating quality and synchronization, and recently, there has been an increasing focus on the importance of intelligibility. Despite these efforts, there remains a challenge in achieving a balance among quality, synchronization, and intelligibility, often resulting in trade-offs that compromise one aspect in favor of another. In light of this, we propose SyncTalklip, a novel dual-tower framework designed to overcome the challenges of synchronization while improving lip-reading performance. To enhance the performance of SyncTalklip in both synchronization and intelligibility, we design AV-SyncNet, a pre-trained multi-task model, aiming to achieve a dual-focus on synchronization and intelligibility. Moreover, we propose a novel cross-modal contrastive learning bringing audio and video closer to enhance synchronization. Experimental results demonstrate that SyncTalklip achieves state-of-the-art performance in quality, intelligibility, and synchronization. Furthermore, extensive experiments have demonstrated our model's generalizability across domains. The code and demo is available at https://sync-talklip.github.io.

## CCS CONCEPTS

• **Computing methodologies** → **Computer vision**; *Phonology / morphology*.

## KEYWORDS

multi-task learning, multimodal learning, talking head generation

## 1 INTRODUCTION

In the last few decades, Talking Face Generation (TFG) has emerged as a key technology in the field of Human-Computer Interaction (HCI) applications [12]. It is used in a range of activities, from movie dubbing [15], face animation [10, 27] to assisting communication for the hearing-impaired through lip-reading. Owing to its broad utility, TFG has attracted increasing attention from both industry and academia.

The goals of the TFG are to generate high-quality, synchronized, and lip-readable videos, which is shown in Fig. 1. The first two of these have been achieved. For temporal synchronization, the

Permission to make digital or hard copies of all or part of this work for personal or classroom use is granted without fee provided that copies are not made or distributed for profit or commercial advantage and that copies bear this notice and the full citation on the first page. Copyrights for components of this work owned by others than the author(s) must be honored. Abstracting with credit is permitted. To copy otherwise, or republish, to post on servers or to redistribute to lists, requires prior specific permission and/or a fee. Request permissions from permissions@acm.org.

*ACM MM, 2024, Melbourne, Australia*

© 2024 Copyright held by the owner/author(s). Publication rights licensed to ACM.
ACM ISBN 978-x-xxxx-xxxx-x/YY/MM
https://doi.org/10.1145/nnnnnnn.nnnnnnn

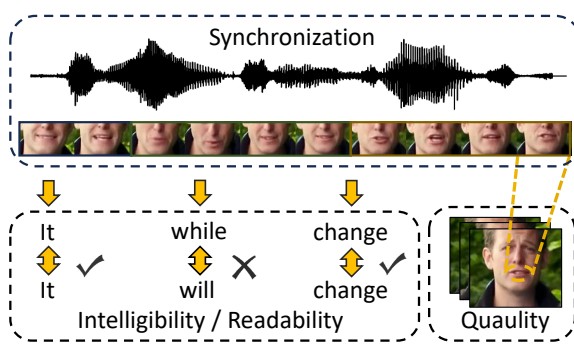

**Figure 1: The overview of the TFG task. Intelligibility is defined as the extent to which the lip movements in the generated video can be comprehended through lip-reading. Synchronization denotes the alignment of audio and video on a temporal scale.**

solution strategy involves integrating an auxiliary network [5] to ensure audio-visual alignment. For generating quality, the strategy involves skip connections [25] and generative adversarial networks (GANs) [3, 9, 39]. The former helps to preserve the temporal dynamics of facial features, while the latter aids in simulating realistic visual output. However, merely achieving high image quality and lip-speech synchronization is not enough to ensure accurate content transmission, because a high-quality image may contain fine-grained lip errors, and precise synchronization may convey incorrect textual content. The McGurk effect [7] points out that when people simultaneously hear and see the semantically mismatched, yet synchronized, sequence of speech and lip movements, they may recognize a phoneme from either the audio or the video, or a fusion of the two. This demonstrates that the audio may mislead you into believing that the generated video is lip-readable. Therefore, to enhance the video's intelligibility, it is essential to detach the audio and separately consider the video's lip-reading performance. Recent work [34] introduces a lip-reading expert to understand content, but sacrifices synchronization and generation quality for intelligibility. Based on this background, our work is dedicated to improving intelligibility while preserving quality and synchronization.

Synchronization can be seen as a temporal-level alignment, while intelligibility can be seen as a semantic-level alignment. To accomplish this dual-level alignment, we propose SyncTalklip, which synthesizes the interaction mechanism between visual and auditory information. This approach ensures that the visual representation is not only closely synchronized with the input speech but also more accurately reflects the true meaning of the linguistic content. To address the issue of audio and video features decoupled during

fine-tuning for the Visual Speech Recognition (VSR) task [26], we design AV-SyncNet. It can better encode semantic-aligned embeddings from audio and video modalities. Besides, We design a novel contrastive learning approach to enhance lip-speech synchronization, which shares a joint embedding space with AV-SyncNet.

Our main contributions are as follows:

- We propose AV-SyncNet to balance synchronization and lip-reading performance, which can better encode semantic-aligned embeddings from audio and video modalities.
- We introduce SyncTalklip, which applies AV-SyncNet as a supervisor and an initialization of the audio encoder. Additionally, SyncTalklip incorporates a novel cross-modal contrastive learning strategy for fine-grained alignment of audio-video embeddings.
- Experimental results demonstrate that SyncTalklip achieves state-of-the-art performance in quality, intelligibility, and synchronization, yielding speaking faces that highly synchronize audio contents while matching the linguistic message.
- We explore a variety of avenues for optimizing the model and share the benchmark codes publicly, thus providing new directions of exploration and a basis for further research.

## 2 RELATED WORK

### 2.1 Talking Face Generation

Initial work [18, 28] used deep learning to learn the mapping of a single speaker's speech to a lip, and then inputting that person's speech into the model to get the corresponding lip. As they are trained on only a specific speaker, they cannot synthesize new identities or voices. However, real-world applications require models that can easily handle generic identities, which means they can accurately generate the corresponding lip movements given any speech. This necessity has led to the development of speaker-independent models [13, 16]. As the need for synchronization and quality becomes increasingly critical, most of the subsequent work focuses on improving realism, audio-lip synchronization, etc., ignoring the intelligibility issue. [4] employs facial keypoints to bridge audio and video, and uses adversarial loss to make the generated video more realistic. [24] enhances synchronization by introducing an additional synchronization discriminator. [40] considers the video's speaking content, head gesture, and identification are considered separately, which contributes in gesture control. In addition, it applies contrast learning ideas to alignment constraints.

In recent work, [34] uses a lip-reading expert to deal with the reading intelligibility problem of the TFG task, introduces the idea of contrast learning to enhance synchronization, and uses [5], a state-of-the-art network in the field of audio-video alignment, for evaluation. However, it can not balance the individual properties well enough to achieve both good synchronization and lip-reading performance without compromising on metrics such as generation quality. So we propose SyncTalklip, which employs a new auxiliary lip-reading network to guarantee lip-reading performance while preserving synchronization in a more reasonable way.

### 2.2 Audio And Video Voice Alignment Learning

AV-HuBERT [26] is a framework for self-supervised representation learning [21] based on audio and video. In the pre-training phase, it can map paired audio-video embeddings close together. Forcing the model to capture temporal relations through a mask prediction task helps to learn the relationship between speech and lip movements, demonstrating that it can align the audio-video features semantically. The fine-tuned model performs well on the lip-reading task. Therefore, supervising the TFG task with AV-HuBERT as a discriminator can lead to an improvement in the intelligibility of the generated video. However, synchronization is not focused on in this self-supervised learning model.

SyncNet [5] is a deep neural network specifically designed for visual synchronization of audio in video. The model trains two independent coding networks for temporal offset detection, which enables the network to recognize temporal deviations between audio and video streams.

AV-HuBERT implements semantic-level alignment and Sync-Net implements temporal-level alignment. Although both are significant for audio-visual content, the focus on both components remains insufficient in existing work. Therefore, we designed AV-SyncNet, a pre-trained multi-task model aiming to integrate the strengths of both.

## 3 METHOD

We propose a SyncTalklip network as shown in Fig. 2. Given our focus on the mouth area, we mask and predict only the lower part of the image to enhance detail capture. AV-SyncNet is introduced as a lip-reading supervisor to obtain intelligibility improvement through lip-reading loss. A novel cross-modality contrastive learning loss is employed to improve synchronization. The model trains audio and video encoders and generators during gradient backpropagation.

### 3.1 Rethinking Talking Face Generation

Although the AV-HuBERT [26] model is highly specialized for lip-reading and the SyncNet [5] model excels in synchronization, neither model fully capitalizes on the combined benefits of these aspects. Previous TFG works [34] have considered the importance of intelligibility of the generated video but have not completely harnessed the potential of synchronization and intelligibility together. Therefore, we introduce SyncTalklip, which uses AV-SyncNet as a submodule and aims to integrate the strengths of these two areas. By finely tuning the balance between lip-reading accuracy and audio-video synchronization, SyncTalklip creates talking faces that are both high-quality, synchronized, and intelligible.

Define $f^a \in \mathbb{R}^{T \times D}$, $f^v \in \mathbb{R}^{T \times D}$ as the features generated by the audio and video encoders, respectively, where T is the sequence length and D is the embedding dimension. Previous work [34] has attempted to directly bring $f^a$ and $f^v$ closer using cosine similarity, which can be expressed as:

$$Sim(f^a, f^v) = \frac{f^a \cdot f^v}{\|f^a\|_2 \cdot \|f^v\|_2} \in (-1, 1) \quad (1)$$

But there is no guarantee that the cosine similarity of the corresponding paired audio-video feature is the maximal. Specifically, denote $(f_i^a, f_i^v)$ as the paired audio-video features, and $\{(f_i^a, f_j^v)\}_{j \neq i}$ as the unpaired audio-video features. There is no guarantee that:

$$\forall j \in \{1, \ldots, T\}, j \neq i \quad Sim(f_i^a, f_i^v) \geq Sim(f_i^a, f_j^v) \quad (2)$$

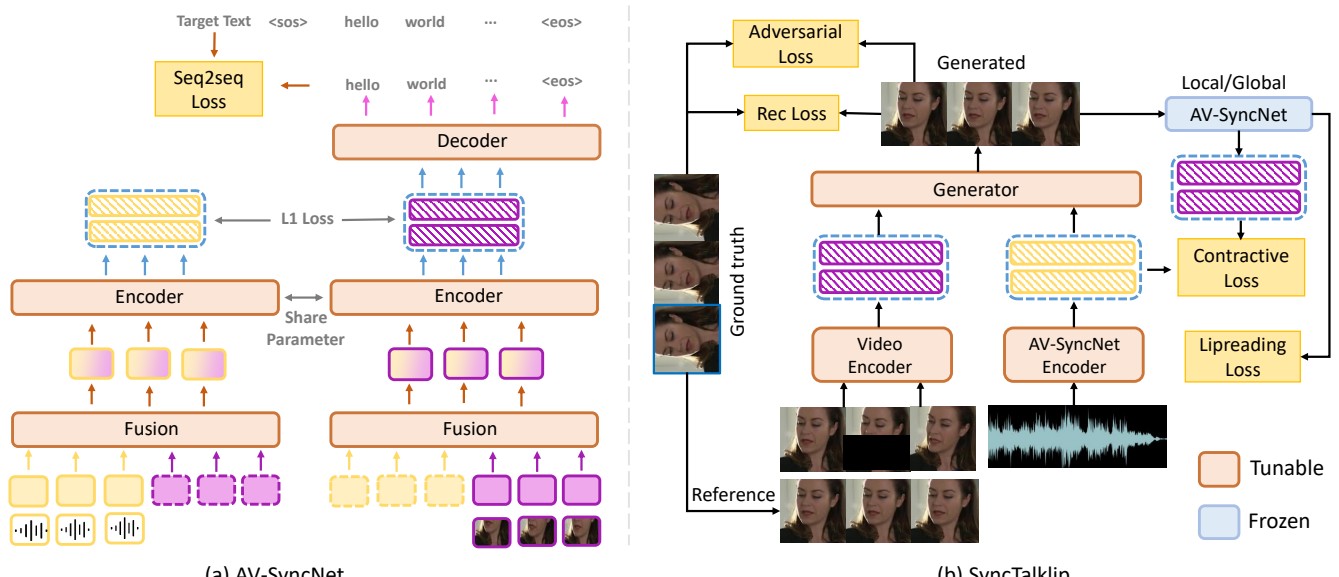

**Figure 2: The overview of our two-stage training strategy. Stage 1, AV-SyncNet is pre-trained for the lip-reading task, simultaneously ensuring fine-grained alignment of the audio-video embeddings. We consider two different methods of alignment: local-level and global-level. Note that the audio is pre-processed by ResNet[29] and the video is pre-processed by FFN[11]. Stage 2, the trained AV-SyncNet is applied to SyncTalklip, so that the generated video combines the performance of both synchronization and intelligibility.**

In fact, there isn't even a guarantee that $Sim(f_i^a, f_i^v)$ will be relatively large among $\{Sim(f_i^a, f_j^v)\}_{j=1}^T$. Our AV-SyncNet model essentially customizes a more sensible distance metric for SyncTalklip, which can be seen as a customized distance abstract space. It ensures that the lip-reading performance degrades as little as possible while bringing the L1 distance of $f^a$ and $f^v$ closer during the training process. Through this metric, the distance of the paired audio-video features, denoted as $Dist(f_i^a, f_i^v)$, will be relatively small among $\{Dist(f_i^a, f_j^v)\}_{j=1}^T$. In this way, when training SyncTalklip, it becomes more effective to draw the audio-video features closer within the same distance abstraction space.

## 3.2 AV-SyncNet

AV-SyncNet merges the benefits of semantic-level alignment and temporal-alignment to avoid subtle lip synchronization problems while ensuring the accuracy of visual representation. SyncTalklip will map paired audio-video embeddings close together during pre-training, but downstream tasks will corrupt the alignment. Therefore, we constrain the audio-video embeddings through L1 loss, which can better converge to a local minimum than other losses [38], such as L2 loss. Drawing closer in this abstract space may also give the model a better understanding of other attributes, such as speech habits and facial expression preferences [32], which will be proved in Sec. 4. Knowledge learned with different goals interacts and reinforces each other, so the process of drawing closer is also a process of gaining a deeper understanding of semantics. In Fig. 2 (a), the encoder is pre-trained and fintune on the downstream

task. Then it is used as an initialization of the audio encoder and a lip-reading expert in the SyncTalklip.

*Pretrain.* The encoder is pre-trained using a self-supervised method, which is the same as [11]. Two steps are alternated during pre-training: feature clustering and mask prediction. The clustering phase uses a discrete latent variable model to form $\{z_t\}_{t=1}^T$, and the model then performs mask prediction through the Transformer [30] architecture to learn a better representation of the audio and video in the semantic space $f^m = \{f_t^m\}_{t=1}^T \in \mathbb{R}^{T \times D}$, where T is the sequence length, D is the embedding dimension, and m means modality. Given the output probability $\{p_t\}_{t=1}^T$, The pre-training loss is:

$$L = -\sum_{t \in M_a \cup M_v} \log p_t(z_t) - \alpha \sum_{t \notin M_a \cup M_v} \log p_t(z_t) \quad (3)$$

where $M^a$ and $M^v$ denote the frames that are masked for the audio and video. $\alpha$ controls the contribution of the unmasked regions in the overall objective.

*Finetune.* The pre-trained model is fine-tuned for the lip-reading task, and the paired audio-video embedding is simultaneously aligned using Eq. 5, which effectively solves the fine-grained synchronization problem, and at the same time maintains the lip-reading ability. As shown in Fig. 2 (a), with audio-only data as input, the data is processed by the fusion module and pre-trained encoder as $f^a = \{f_t^a\}_{t=1}^T \in \mathbb{R}^{T \times D}$. Similarly, the video-only data is processed as $f^v = \{f_t^v\}_{t=1}^T \in \mathbb{R}^{T \times D}$. The $L_{av}$ is then used to

bring $f^a$ and $f^v$ closer under L1 distance. A tunable transformer decoder is appended to autoregressively decode $f^v$ into probabilities $p(\omega_t|\{\omega_i\}_{i=1}^{t-1}, f^v)$, where $\omega_i$ is the ground-truth transcription.

*Lip-reading Loss.* The lip-reading loss is used to maintain the intelligibility of the model, which is a sequence-to-sequence loss. It is calculated after the decoder module using cross-entropy loss. Define $S$ as the length of the target text, and the lip-reading loss can be expressed as:

$$L_{lip} = -\sum_{t=1}^{S} \log p(w_t|\{w_i\}_{i=1}^{t-1}, f^v). \tag{4}$$

*Av-alignment Loss.* Considering the problem mentioned in Sec. 3.1, this loss is used to create a customized distance metric for SyncTalklip, which is a crucial basis for Eq. 12. Define $\tau$ as a temperature ratio, which is a hyperparameter that weights the degree of synchronization. Then the $L_{av}$ can be expressed as:

$$L_{av} = \mathcal{F}\left(\|f^a - f^v\|_1 \cdot \tau\right) \tag{5}$$

Define $L_{lip}$ as the loss of the lip-reading, and $L_{av}$ as the loss of bringing embeddings closer. The AV-SyncNet loss is:

$$L_{total} = \lambda_{av} \cdot L_{av} + \lambda_{lip} \cdot L_{lip} \tag{6}$$

where $\lambda_{av}$ and $\lambda_{lip}$ scale the contributions of different loss items.

## 3.3 SyncTalklip

As shown in Fig. 2 (b), the generated frames first undergo frame-level loss computation, after which the predicted frames are added back to the original sequence for stream-level loss computation. Denote the original video frame as $v_i \in \mathbb{R}^{C \times H \times W}$, and the generated video frame as $\hat{v}_i \in \mathbb{R}^{C \times H \times W}$, where $C$ denotes the channels. Take the prediction of frame $i$ as an example, the complete sequence is $\hat{V} = [v_1, ..., \hat{v}_i, ..., v_T] \in \mathbb{R}^{T \times C \times H \times W}$. The encoded modality embeddings $f^a$ and $f^v$ are fed into the generator, which yields $\hat{v}_i$. Then compute the frame-level loss of $(v_i, \hat{v}_i)$: reconstruction loss and adversarial loss. Subsequently, the lip-reading effect of the generated video stream $\hat{V}$ is evaluated with a frozen AV-SyncNet, from where the lip-reading loss is calculated. At the same time, it can also generate the video embedding $f^v$ as a byproduct. The tunable audio encoder is init by the AV-SyncNet encoder, and generates the audio embedding $f^a$. Then $f^a$ and $f^v$ are brought closer through our novel cross-modality contrastive loss. Stream-level loss including contrastive loss and lipreading loss controls the generation ability, affecting the semantic and temporal alignment of the generated audio-visual outcome. In this process, all gradients are back-propagated to the generative system and the discriminator.

*Reconstruction Loss.* Reconstruction loss is used to minimize the distance between the generated image and the real image. In this way, the model learns the pixel-level details of the generated image. The reconstruction loss is formulated as:

$$L_{rec} = \frac{1}{T} \sum_{i=1}^{T} \|v_i - \hat{v}_i\|_1 \tag{7}$$

*Adversarial Loss.* Adversarial Loss utilizes the framework of Generative Adversarial Networks (GAN) [6], this loss function forces the generated images to be more visually realistic. Through adversarial training [14], the generator learns to create realistic-looking images that are difficult to distinguish by the discriminator, thus enhancing the realism of the generated images. $v$ represents a sample drawn from a real data distribution, and $\hat{v}$ is the generated frames by our model. $D(v)$ denotes the probability that the discriminator network evaluates $v$ to be true. The discriminator $D$ tries to output a high probability value for the true sample $v$ and a low probability value for the fake sample generated by SyncTalklip. The model minimizes $L_{gan}$:

$$L_{gen} = \mathbb{E}[\log(1 - D(\hat{v}))] \tag{8}$$

$$L_{disc} = \mathbb{E}[\log(1 - D(v))] + \mathbb{E}[\log(D(\hat{v}))] \tag{9}$$

$$L_{gan} = L_{gen} + L_{disc} \tag{10}$$

*Lip-reading Loss.* Lip-reading loss is generated by the lip-reading supervisor. This loss aims to improve the performance of the system in terms of intelligibility to ensure the correct delivery of linguistic content. It is calculated in the same way as Eq. 4. However, the AV-SyncNet here is frozen, and the gradient is used to update the generation system during backpropagation.

*Contrastive Loss.* Utilizing the concept of contrastive learning, we propose a novel cross-modal contrastive learning strategy for audio-visual fine-grained alignment. Although it differs from traditional methods, the underlying idea remains similar. This strategy enhances the model's performance in synchronization and intelligibility by bringing paired embeddings closer together and increasing the distance between unpaired embeddings in L1 space, which corresponds to AV-SyncNet. Accordingly, the model minimizes the following loss:

$$\theta(x, x') = \mathcal{F}\left(\|f^x - f^{x'}\|_1 \cdot \tau\right) \tag{11}$$

$$L_{cav} = \sum_{i \in \Upsilon} \theta(a_i, v_i) - \frac{\sum_{j \in \Upsilon, j \neq i} \theta(a_i, v_j)}{|\Upsilon| - 1} \tag{12}$$

where the temperature coefficient, denoted by $\tau$, is used to control the degree of scaling, and $f^x$ is the embedding of the modality frame $x$. The symbol $\Upsilon$ represents all masked frames. As mentioned earlier in Sec. 3.3, $\tau = \{i\}$ in that case. The cardinality of the set $\Upsilon$ is given by $|\Upsilon|$. $\mathcal{F}$ represents a linear layer. Since it has been guaranteed by AV-SyncNet that the $Dist(f_i^a, f_i^v)$ will be relatively small among $\{Dist(f_i^a, f_j^v)\}_{j=1}^{T}$, as mentioned in Sec. 3.1, so it is sensible to minimize $L_{cav}$.

By combining different loss functions, SyncTalklip employs a multi-task learning strategy [14] designed to optimize multiple objectives simultaneously. This strategy allows the model to not only perform well on a single task, but also to ensure synchronization of speech and lip movements as well as correct understanding of content while maintaining visual fidelity.

The SyncTalklip loss is:

$$L_{total} = \lambda_{cav} \cdot L_{cav} + \lambda_{gan} \cdot L_{gan} + \lambda_{rec} \cdot L_{rec} + \lambda_{lip} \cdot L_{lip} \tag{13}$$

where $\lambda_{cav}, \lambda_{gan}, \lambda_{rec}$ and $\lambda_{lip}$ scale the contributions of different loss items.

**Table 1: Evaluation on LRS2 and LRS3. Note that the SyncTalklip and TalkLip models are trained on the LRS2 (224h) dataset. $WER_1(\%)$ and $WER_2(\%)$ are scored by the AV-HuBERT, which are train on LRS2 and LRS3 seperately. Some models focus on other features, like 3D, so we didn't compare them in depth.**

| Method | LES2 | | | | | LRS3 | | | | |
|---|---|---|---|---|---|---|---|---|---|---|
| | LSE-C↑ | LSE-D↓ | PSNR↑ | SSIM↑ | $WER_1$ ↓ | LSE-C↑ | LSE-D↓ | PSNR↑ | SSIM↑ | $WER_2$ ↓ |
| Ground Truth | 8.25 | 6.25 | - | 1.00 | 25.0 | 7.62 | 6.88 | - | 1.00 | 28.6 |
| ATVGnet[40]* | 5.05 | 8.65 | 30.42 | 0.75 | 113.7 | - | - | - | - | - |
| 3D Identity Mem[37]* | 4.18 | 9.26 | 29.86 | 0.69 | - | - | - | - | - | - |
| PC-AVS[40]* | 6.73 | 7.30 | 29.89 | 0.74 | - | - | - | - | - | - |
| MakeItTalk[41] | 6.47 | 7.69 | 29.87 | 0.71 | 101.3 | 6.46 | 7.72 | 29.85 | 0.69 | 111.2 |
| Faceformer[8] | 6.42 | 7.80 | 29.47 | 0.84 | 97.6 | 6.44 | 7.71 | 29.79 | 0.69 | 114.7 |
| Wav2Lip[24] | 8.40 | 6.58 | 31.36 | 0.85 | 82.1 | 7.71 | 6.89 | 32.47 | 0.90 | 109.0 |
| TalkLip[34] | 8.53 | 5.87 | 30.42 | 0.81 | 23.4 | 7.97 | 6.59 | 30.21 | 0.80 | 23.2 |
| SyncTalklip (ours) | **9.29** | **5.52** | **31.85** | **0.93** | **19.7** | **8.38** | **6.41** | **32.16** | **0.93** | **19.1** |

Performances of methods with * are collected from [23] and [34]. For [8] and [41], we used the model given in the paper and did a partial test in LRS3.

## 3.4 Trade off between Synchronization and Intelligibility

During the fine-tuning of AV-SyncNet, we adopted a novel strategy that incorporates a convolutional module before the alignment of audio-visual embeddings. This approach is designed to capture deep semantic relationships between neighboring frames, thereby shifting the model's focus from the local level to the global level [17]. However, it comes at the cost of sacrificing fine-grained supervision, in exchange for a deeper comprehension of the semantic information.

To further improve the performance of the model, we freeze the AV-SyncNet audio encoder during the training of SyncTalklip. In this method, the output from the audio encoder serves as the input for computing contrast loss, yet it does not participate in the gradient updates. Instead, an additional audio encoder is employed to participate in the update process. The processing flow of the module is shown in the supplementary in detail.

## 4 EXPERIMENT

### 4.1 Dataset

The LRS3 dataset [2] consists of obtained from TED talks covering a large number of speakers and background noise environments. These videos contain sentences in the English language, which is beneficial for lip-reading and visual speech recognition research. The LRS3 dataset is widely used in research on lip-reading techniques due to its diversity and size. The LRS2 dataset [1] is from BBC, which consists of video, audio, and text for each sample, where the sample rates are 16 kHz for audio and 25 fps for video. The dataset contains more than 1,000 speakers, nearly 150,000 utterance instances, and nearly 63,000 different words, which makes the dataset extremely rich in data. The LRS2 dataset is particularly suitable for studying how to perform effective lip-reading in long video sequences, as it covers continuous natural conversations. The VoxCeleb2 dataset [22] contains over 1 million video clips from YouTube videos. These clips are from over 6,000 celebrities and cover multiple languages, accents, and background noise conditions. VoxCeleb2 is often used to train cross-modal recognition systems, especially those that combine visual and audio information. We only use the English portion of the VoxCeleb2.

### 4.2 Implement Details

We detect faces in each image as the Region of Interest (ROI) and then crop the ROI to $96 \times 96$ pixels. Audio waveforms are preprocessed into mel-spectrograms. For AV-SyncNet, the inputs are lip ROIs for the visual stream and log filterbank energy features for the audio stream. For SyncTalklip, a randomly picked image within the video is used as the identity reference, and the pose reference is the same as the target face image, except that it masks the lower half of the face to prevent our SyncTalklip model from learning movements in the lip region. Based on the pre-trained model, we fine-tune the AV-SyncNet for 180K steps and train SyncTalklip for 60K steps on a single Nvidia 3090 GPU.

AV-SyncNet is pre-trained on LRS3 (433h) and VoxCeleb2, and fine-tuned on LRS2 (224h). The model consists of 24 transformer blocks, 1024 embedding dimensions, 4096 feed-forward dimensions, and 16 attention heads. In the pre-training phase, five clustering-mask iterations are conducted as mentioned in Sec. 3.2. In the fine-tuning phase, the model is guided by a multi-task loss function, which includes lip-reading and local-level alignment tasks. For SyncTalklip (conv) model, a convolutional submodule is added for global-level alignment, which can better capture the temporal knowledge and add flexibility to the overall system.

SyncTalklip is trained on the LRS2 dataset. The training process is conducted as follows: First, in the video part, the predicted video frames are masked and then fed into the audio and video encoders to generate the corresponding audio and video embeddings respectively. The Generation module generates the predicted frames and calculates the frame-level loss and stream-level loss. On the one

**Table 2: Lip-reading effect of AV-SyncNet. AV-SyncNet(conv) denotes the convolution operation applied before calculating the audio and video loss. AV-SyncNet retains a good lip-reading capability while aligning the audio and video, and the alignment quantization results are shown in the supplementary.**

| Method | Train | AV-Align | WER(%)↓ |
|---|---|---|---|
| AV-HuBERT | LRS2(224h) | ✗ | 25.0 |
| AV-SyncNet | LRS2(224h) | ✓ | 26.0 |
| AV-SyncNet(conv) | LRS2(224h) | ✓ | 26.1 |
| Conformer | LRS2+LRS3 | ✗ | 40.9 |

hand, the generated frames are compared with the corresponding frames of ground truth to compute the adversarial loss and reconstruction loss. On the other hand, it is inserted back into the original video sequence. The generated video stream is input to the trained expert system AV-SyncNet for lip reading task to compute the loss, while in the lip reading task a byproduct is generated, i.e., the corresponding video embedding. Contrastive loss is computed by combining the video embedding here with the audio embedding generated during the generation process. The detailed information of our method can be found in the supplementary.

## 4.3 Metrics

PSNR [31] and SSIM [19, 35] are used to evaluate the quality of the generated frames, LSE-D and LSE-C [24] are used to evaluate the synchronization of audio and video, and WER is used to evaluate the performance of the generated video in the lip reading task.

*Quality.* PSNR and SSIM are used to evaluate the image quality, PSNR is based on the average of the squares of the errors and reflects the magnitude of the difference between the original image and the distorted image, the higher the PSNR the better the quality of the image, SSIM takes into account the structural information of the image, the brightness, the contrast, etc., and the closer its value is to 1 means the higher the quality.

*Synchronization.* The LSE-D has been widely used as a measure of synchronization and employs the mechanism of the sliding window in a geometrical sense, which is a minimum of the Euclidean distance from the audio to the center of the corresponding video window [24]. LSE-C is widely used in previous works [24, 34].

*Intelligibility.* The word error rate (WER) is used as the evaluation index of speech recognition, which is defined as $WER = (S + K + I)/N$, where $S$ denotes the number of words replaced, $K$ denotes the number of words deleted, $I$ denotes the number of words inserted, and $N$ denotes the total number of words in the reference text. $WER_1$ and $WER_2$ are scored by the AV-HuBERT (large) [26], which are train on LRS2 and LRS3 seperately, while $WER_3$ is scored by the Conformer [20, 33].

**Table 3: Ablation experiments. w/o novel-contrastive loss: traditional contrastive learning is used instead. w/o AV-SyncNet: AV-HuBERT is used instead. More details is shown in supplementary.**

| Method | LSE-C↑ | PSNR↑ | WER(%)↓ |
|---|---|---|---|
| SyncTalklip | **8.81** | **31.85** | **26.8** |
| w/o novel-contrastive learning | 7.74 | 30.17 | 35.1 |
| w/o AV-SyncNet | 8.53 | 30.42 | 30.4 |
| w/o contrastive loss module | 6.42 | 30.12 | 48.3 |
| w/o lip-reading loss module | 8.40 | 31.36 | 73.9 |

## 4.4 Main Results

We finetune the AV-SyncNet on LRS2, the performance of which is shown in Tab. 2. This result shows that it has good retention in lip-reading ability. The trained model is then used in SyncTalklip. In order to verify the reading intelligibility of the generated videos, the generated videos are put to test the lip-reading task on the LRS2 test set using AV-HuBERT to calculate the Word Error Rate (WER). As illustrated in Tab. 1, SyncTalklip demonstrates excellent performance across various key metrics, particularly in LSE-C, LSE-D, and WER. Specifically, SyncTalklip achieves leading LSE-C and LSE-D scores, which proves its superior synchronization capabilities. The PSNR and SSIM metrics are on par with Wav2Lip, which is well-known for its synchronization capabilities, and the scores of our model are closer to the ground truth.

Moreover, SyncTalklip achieves the lowest WER among all competing methods when evaluated by AV-HuBERT, which achieves SOTA performance on the lip-reading task. Note that $WER_1$ and $WER_2$ for some methods exceed 100%. This is because the evaluator predicts a much longer sentence. To distinguish between the lip-reading supervisor in the model and the evaluator used to assess the intelligibility of the generated images, we utilize the Conformer [33] [20] architecture in teacher-forcing mode[36] for assessment, and the results are shown in Tab. 4. These results underscore the ability of SyncTalklip to produce high-quality visual content without sacrificing synchronization. Overall, these results underscore the superiority of SyncTalklip in TFG.

## 4.5 Ablation

As shown in Tab. 3, a series of experiments are conducted to verify the usefulness of the lip-reading module, novel contrastive learning module, and AV-SyncNet. The module ensuring generation quality has been validated for its efficacy [24, 34]. Therefore, we will subsequently conduct ablation studies solely on the modules responsible for synchronization and intelligibility.

*Ablation on novel contrastive learning module.* Using AV-SyncNet model as the supervisor, the contrastive learning module is removed, while the lip-reading module is retained. The experimental results reveal that the videos generated by the model show a significant dip in synchronization and a slight decrease in intelligibility. Furthermore, we still use AV-SyncNet model as the supervisor, but employ traditional contrastive learning instead, and the lip-reading module is retained. The experimental results reveal that the videos

**Table 4: Model Optimization Exploration. The second row displays the effects of adding a convolution module, while the third row demonstrates the effects of freezing the AV-SyncNet module. The right side of the table illustrates the results of the generalizability test conducted on the LRS3 dataset. WER$_3$(%) is scored by Conformer.**

| Method | LRS2 | | | | | LRS3 | | | | |
|---|---|---|---|---|---|---|---|---|---|---|
| | LSE-C↑ | LSE-D↓ | PSNR↑ | SSIM↑ | WER$_3$ ↓ | LSE-C↑ | LSE-D↓ | PSNR↑ | SSIM↑ | WER$_3$ ↓ |
| TalkLip | 8.53 | 5.87 | 30.42 | 0.81 | 30.4 | 7.97 | 6.59 | 30.21 | 0.80 | 30.4 |
| SyncTalklip | **9.29** | **5.52** | 31.85 | **0.93** | 26.8 | **8.38** | **6.41** | 32.16 | 0.93 | **42.5** |
| SyncTalklip(conv) | 9.16 | 5.58 | 30.79 | 0.92 | **25.6** | 8.27 | 6.50 | 30.98 | 0.92 | 44.3 |
| SyncTalklip(freeze) | 8.21 | 6.35 | **31.97** | **0.93** | **25.6** | 7.29 | 7.24 | **32.20** | **0.94** | 81.8 |

generated by the model show a slight decrease in both synchronization and intelligibility. These two experiments demonstrate the contribution of our novel cross-modality contrastive learning strategy.

*Ablation on lip-reading module.* Using AV-SyncNet model as the supervisor, the lip-reading module is removed, while the novel contrastive learning module is retained. It shows that this configuration of the model achieves good results in maintaining synchronization and quality. However, it is also worth noting that the model exhibits significant shortcomings in terms of intelligibility.

*Ablation on AV-SyncNet.* Using the AV-HuBERT model as the supervisor, the novel contrastive learning module and lip-reading module are retained. It shows that the videos generated by the model have a slight decrease in all attributes, which proves the significance of AV-SyncNet.

These experimental results also demonstrate that relying solely on either module fails to yield the desired outcomes. This highlights the critical need for integrating these modules to produce video content that is high-quality, synchronized, and comprehensible.

## 4.6 Appropriate Model is All You Need

To trade off between synchronization and intelligibility, we conducted convolution and freeze exploration. As shown in Tab. 4, the models outperform in different aspects. We also performed zero-shot generalizability tests on the LRS3 dataset using the model trained on LRS2.

*Convolution.* During the fine-tuning of AV-SyncNet, we explored a strategy that involves incorporating a convolutional module before computing the distance between vectors. After extensive training, our model has optimized the Word Error Rate (WER) by 1.2% compared to the baseline, as detailed in Tab. 4. However, this improvement in intelligibility comes at the expense of a slight decrease in synchronization and authenticity. The convolutional process shifts the model's focus from local to global features. A larger convolution kernel enables the model to capture more holistic semantic information, prioritizing a deeper understanding of the content over fine-grained details. Conversely, a smaller kernel preserves local nuances, enhancing the quality of individual frames. A larger window size offers profound insights into the overall semantics, but this can result in a lack of detailed one-to-one supervision and lower the single-frame quality. Fundamentally, this represents a

balance between a global perspective and local precision within the architecture of the model.

*Freeze.* To further improve the model's performance, we experimented with freezing the AV-SyncNet audio encoder during the training of SyncTalklip. The results are shown in Tab. 4. The experimental results demonstrate that the frozen SyncTalklip achieves a 1.2% improvement in word error rate (WER) compared to the base model, indicating an increase in content comprehensibility. However, there is a noticeable decrease in synchronization. This outcome can be attributed to the freezing operation, which preserves the lip-reading performance of AV-SyncNet, thereby enhancing its capability in this specific aspect. Nevertheless, this preservation also constrains the potential for synchronization improvements due to insufficient supervision of the audio-video synchronization. Additionally, after freezing, the model shifts a greater focus onto Generative Adversarial Networks (GANs), leading to further enhancements in the quality of the generated images. This strategy produces results that excel in terms of intelligibility and generation quality, making it meaningful for improving human-computer interaction experiences in dynamic real-world scenarios.

*Generalizability on the LRS3.* In addition, we evaluated the generalization ability of the models on the LRS3 dataset, which is shown in Tab. 4. Compared to SyncTalklip (freeze), SyncTalklip and SyncTalklip (conv) are more capable of generalization, which means that using the SyncTalklip (freeze) model on zero-shot datasets should be done with caution. The freeze operation inherently limits the number of parameters that can be updated, thus reducing the ability of the model to absorb new information. This reduction in model complexity affects its flexibility, making it less adaptable than its unfrozen counterpart. Interestingly, the freeze operation seems to focus the model's learning on improving the quality of image generation.

## 4.7 Perceptual Evaluation

One hundred videos, randomly selected from real and generated datasets, are evaluated in our study. Each video is assessed by one participant, with a total of twenty participants involved. The scoring range is from one to five. The evaluation covered three main dimensions: video intelligibility, i.e., how clearly the viewer can understand the verbal content in the video; authenticity, which assesses how similar the generated video is to a real human video; and synchronization, i.e., how well the lip movements in the video

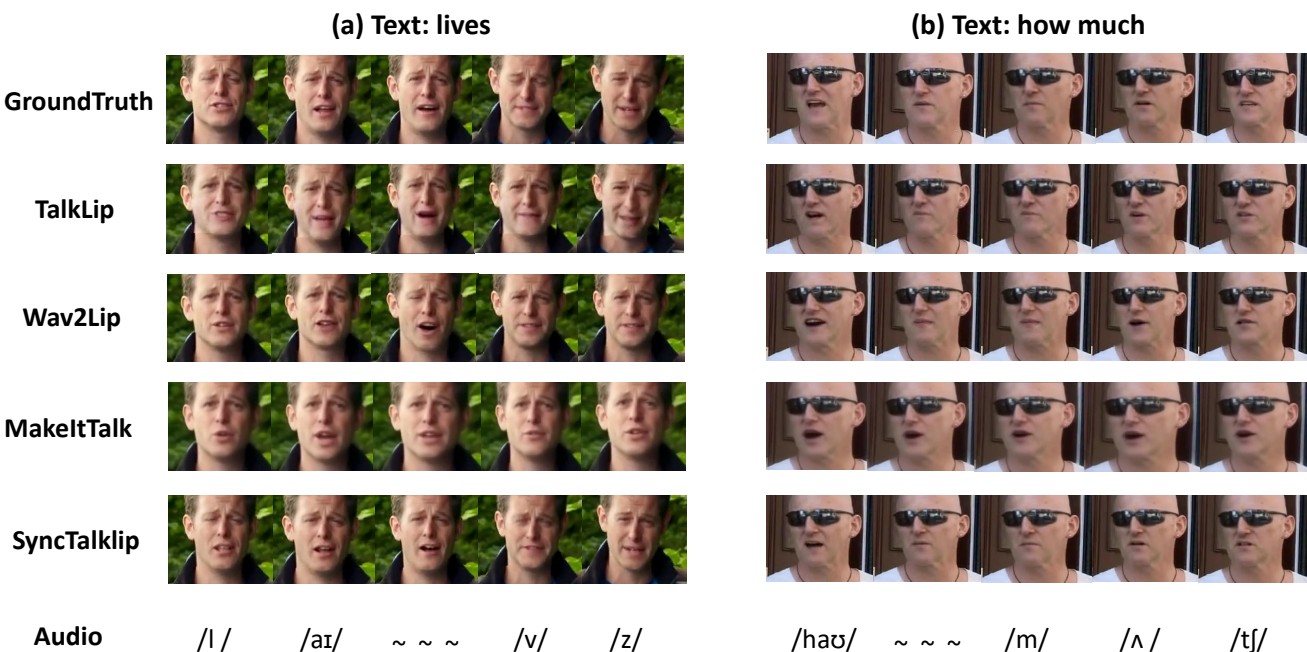

**Figure 3: We compare sequential snapshots from two video-generated sentences to evaluate various methods. The top row, denoted as Ground Truth, depicts the actual lip movements from the video. Below, the synthesized frames from the models SyncTalklip, TalkLip, and Wav2Lip are presented. Our method closely mirrors the ground truth, particularly in lip synchronization and the intelligence of mouth shapes, demonstrating superior performance in both video quality and the realism of mouth movements. More examples can be seen in supplementary.**

**Table 5: Perceptual Evaluation. INTL, SYNC, and AUTH stand for Intelligibility, Synchronization, and Authenticity, respectively.**

| Method | INTL↑ | SYNC↑ | AUTH↑ |
|--------|-------|-------|-------|
| Wav2Lip | 2.43 ± 1.12 | **4.54 ± 0.47** | 4.35 ± 0.22 |
| TalkLip | 3.12 ± 1.24 | 3.84 ± 0.72 | 3.25 ± 0.47 |
| SyncTalklip | **3.27 ± 0.76** | 4.44 ± 0.43 | **4.41 ± 0.23** |
| Ground Truth | 3.53 ± 0.93 | 4.85 ± 0.14 | 4.96 ± 0.04 |

match the speech. During the evaluation process, we asked the evaluators to score the videos based on these three dimensions after watching each video sample, so as to comprehensively assess the quality of the videos generated by our method and its performance in different aspects. The experimental results show that our model is superior in perceptual evaluation, which is shown in Tab. 5.

In order to qualitatively evaluate the different approaches, we show two snapshots of the generated speech face videos in Fig. 3. These snapshots visualize the generation quality of each model, and we encourage to go to the demo page to see a more detailed presentation. Specifically, the first row provides snapshots of the real videos, followed by images synthesized by various methods. Through the figures, we are able to see that the image frames

generated by our proposed method are very similar to the real video and perform the best in terms of video quality and naturalness of mouth movements as well.

## 5 CONCLUSION

In this paper, we analyze the challenges faced in the TFG task, and propose a novel SyncTalklip network to synthesize talking face videos with great quality, synchronization, and reading intelligibility. The Experimental results indicate that SyncTalklip surpasses existing methods across all the key performance metrics. A novel cross-modality contrastive learning approach is adopted in our model, which mainly contributes to the synchronization. Besides, AV-SyncNet customized a distance metric for SyncTalklip, whose effectiveness has been demonstrated. Additionally, we introduce variants such as global-level SyncTalklip and AV-SyncNet-frozen SyncTalklip, which excel at different attributes. Based on your needs, you can choose the appropriate model among the three. Moving forward, we aim to further explore audio-video synchronization and content generation technologies to realize more natural and enriched human-computer interaction experiences. Meanwhile, we anticipate that the community will engage in more extensive testing and application of SyncTalklip, thereby driving progress and fostering innovation in related technologies.

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
