# OpenReview forum: "SyncTalklip: Highly Synchronized Lip-Readable Speaker Generation with Multi-Task Learning"
_acmmm.org/ACMMM/2024/Conference — MM2024 Poster_

### Official Review · Reviewer_Nzg9 · 2024-05-16

**Rating:** 1
**Confidence:** 4

**Summary:**

This paper proposes to use AV-SyncNet to enhance both synchronization and intelligibility and analyzes the challenges faced in the TFG task. The authors also propose a cross-modal contrastive learning bringing audio and video closer to enhance synchronization. The experiments are conducted on LRS2 and LRS3 to validate the effectiveness of the proposed method.

**Strengths:**

The paper proposes an auxiliary model (AV-SyncNet) to improve TFG performance, refining a contrastive loss to minimize paired audio-visual feature distance.

**Limitations:**

This paper exhibits significant limitations in terms of problem definition, motivation, and experiments, which are outlined as follows:

1. This paper claims “Recent work [34] introduces a lip-reading expert to understand content, but sacrifices synchronization and generation quality for intelligibility.” in Line 103-105 and “Previous TFG works [34] …” in Line 210-212. However, the reported results from TalkLip [34] indicate that while they improved intelligibility, they also improved synchronization, which conflicts with the authors' description.

2. TalkLip employs AV-Hubert as an auxiliary model to improve intelligibility and synchronization, but it leads to a decline in quality. This paper utilizes a similar model AV-SyncNet and manages to enhance all three aspects simultaneously. In addition to explaining the improvements in intelligibility and synchronization, it should also clarify why quality is also enhanced.

3. In Line 220, the authors claim previous work [34] used cosine similarity to make the audio and video features closer. However, TalkLip [34] employs an infoNCE-based contrastive learning strategy to align audio and video features, rather than cosine similarity.

4. The reported results in Table 1 of the paper are questionable, as the authors mention utilizing the models from the papers (FaceFormer [8] and MakeItTalk [41]) for testing. However, the results for FaceFormer and Wav2Lip on the LRS2 dataset seem simply copy from those reported in TalkLip [34].

5. The Table 1 states that TalkLip was trained on LRS2 (224h), but to my knowledge, TalkLip was trained on LRS2 (29h). Additionally, what does LES2 refer to in the table?

6. How is the dataset divided?

7. The author should include comparisons with commonly used person-generic TFG methods, such as VideoReTalking and IP-LAP.

8. In the paper and supplementary, I cannot find the visualization results for FaceFormer, which is inconsistent with the quantitative results.

9. The comparison in Figure 3 is unfair, for example, as can be seen in the last column of the right figure, there is no short white vertical bar in the bottom left corner of GT, TalkLip, and MakeItTalk, while it appears in wav2Lip and the proposed method.

10. The caption of Figure 1 aims to describe an overview of the TFG task, but the content of the figure does not reflect the process of the TFG task. Instead, it seems more like a description of performance evaluation metrics.

11. "talking face generation" might be a more suitable keyword than "talking head generation," as the latter term does not appear in the paper.

12. There are some issues with the expression of the formulas, such as inconsistencies between the symbols in Formula 3 and their corresponding explanations, lack of explanations for the symbols in Formula 5, and the use of the same symbol ($L_{total}$) in Formulas 6 and 13.

13. The English usage needs to be improved, for instance, the misspelled 'fintune' on Line 289.

**Suitability:**

3

---

### Official Review · Reviewer_Rwm2 · 2024-05-21

**Rating:** 4
**Confidence:** 3

**Summary:**

The paper introduces SyncTalklip, a novel framework designed to generate talking face videos that achieve a balance between visual quality, synchronization with audio, and lip readability. This is achieved by leveraging a dual-tower framework and a multi-task learning approach. The core innovation is the use of AV-SyncNet, a pre-trained model that facilitates both synchronization and intelligibility, along with a novel cross-modal contrastive learning strategy.

* AV-SyncNet: A pre-trained multi-task model that integrates audio and video synchronization with lip-reading performance.
* SyncTalklip Framework: Incorporates AV-SyncNet as a supervisor and an initialization of the audio encoder, enhancing synchronization and lip-reading accuracy.
* Cross-Modal Contrastive Learning: A novel strategy to bring audio and video embeddings closer, improving synchronization without sacrificing intelligibility.
* Experimental Validation: Demonstrates state-of-the-art performance in quality, intelligibility, and synchronization through extensive experiments.

**Strengths:**

* Novel Framework: The introduction of SyncTalklip addresses a significant gap in the balance between synchronization, quality, and intelligibility in talking face generation.
* Dual-Tower Design: Effective integration of AV-SyncNet ensures robust synchronization and intelligibility.
* Innovative Learning Strategy: The cross-modal contrastive learning approach is a novel contribution that enhances the model's performance in aligning audio and video embeddings.
* Comprehensive Evaluation: The model is extensively tested on multiple datasets (LRS2, LRS3, VoxCeleb2) and shows superior performance compared to existing methods.

**Limitations:**

* Complexity: The dual-tower framework and the multi-task learning approach add complexity to the model, which may make it challenging to implement and require significant computational resources.
* Generalization: While the model shows strong performance on specific datasets, its generalization to entirely new datasets or real-world scenarios may require further validation.
* Limited Focus on Fine-Grained Details: The shift towards global semantic information might result in the loss of some fine-grained details, affecting the overall quality in certain scenarios.

**Suitability:**

3

---

### Official Review · Reviewer_seZr · 2024-05-24

**Rating:** 4
**Confidence:** 4

**Summary:**

This paper introduces a novel framework designed to address the challenges of generating high-quality, synchronized, and intelligible talking face videos. The authors highlight the difficulties in balancing quality, synchronization, and intelligibility in existing Talking Face Generation (TFG) systems, which often lead to trade-offs among these aspects.

**Strengths:**

The proposed SyncTalklip framework incorporates a dual-tower architecture that focuses on synchronization and intelligibility. It utilizes a pre-trained multi-task model called AV-SyncNet, which is designed to encode semantic-aligned embeddings from both audio and video modalities. A key innovation of SyncTalklip is the introduction of a novel cross-modal contrastive learning approach that enhances lip-speech synchronization.

The experiments and ablation studies are extensive and representative.

**Limitations:**

1.The statement of “This approach ensures that the visual representation is not only closely synchronized with the input speech but also more accurately reflects the true meaning of the linguistic content.” is not true. When the audio signal is clean (without background noise, ASR is more accurate than VSR).
2.Talklip [34] also uses the contrastive strategy to enhance lip-speech synchronization by attracting audio embeddings and their time-aligned visual context features while repelling the audio embeddings from different frames. This functionality is very similar to the proposed AV-SyncNet. Thus, I think the authors should better clarify their difference to TalkLip and refine the first contribution point.

**Suitability:**

3

---

### Official Review · Reviewer_tE5w · 2024-05-25

**Rating:** 4
**Confidence:** 3

**Summary:**

The paper address the problem of talking-face generation. The major claim of the paper is to enhance the intelligibility of the model, which is a major drawback in most of the existing talking-face generation works. The idea of using AV-Hubert as an additional cue to understand the linguistic content is interesting and practical. The quantitative results indicate good performance in comparison to existing works on two standard talking-face datasets, however the qualitative results are limited in number and also on the variety of samples shown.

**Strengths:**

- The idea of using AV-Hubert as a supervisor for improving the intelligibity is interesting. The strategy of adding lip-reading loss is simple, yet effective and stronly imporves the performance of the network (in terms of WER).
- The paper is well written and easy to follow.
- Quantitative results are impressive, specifically WER. While the existing models achieve high quality and good synchornization, the ability to understand the underlying content is relatively lower, which is tackled in this work.

**Limitations:**

- Unfreezing AV-SyncNet for lip-reading loss: Authors mention that the AV-SyncNet is frozen to compute the lip-reading loss in SyncTalkLip. What happens if AV-SyncNet is unfrezzed? Does this lead to additional improvements in the performance or it only adds to the overhead in the training without significant gains?
- Qualitative results:
    - The results presented in the demo page are limited. To better understand the capabilities of the model, it would be good to inlcude more qualitative samples.
    - Longer samples needed - The current results are very short, now when the major claim of the paper is on intelligibility, it becomes important to judge how the model performs in generating longer sentences as opposed to short phrases like shown currently.
    - More results on real-world examples - The current results shown are from the datasets. However, the real applicability of the model lies in its ability tp perform on real, in-the-wild samples. It would be good to include more such results.
- Improvement in visual quality: The current results from the model are for very low-resolution, 96x96 pixels (can also be seen in qualitative results). However, for the model to be truely application in real settings, it need to perform well on sufficiently higher resolutions. It would be beneficial to experiment on some higher resolution videos and analyse the performance.

- Minor comments:
    - Typos:
        - Line 119 - Besides, We design a novel -> No need to capitalalize **W** in "We"
        - Line 289 -  fintune on the downstream ->  **fintuned** on the downstream
    - Improvement in Fig.2 - The cuurent diagram is a bit confusing and hard to understand, the authors can consider re-designing the diagram to make the flow easier.

**Suitability:**

3

---

### Meta-Review · Area_Chair_MhNE · 2024-06-30

**Recommendation:** Accept (Poster)
**Confidence:** 4

**Metareview:**

The authors discuss lip-readable speaker generation using multi-task learning. However, the proposed AV-SyncNet is similar to TalkLip [34], which used a contrastive strategy to enhance lip-speech synchronization by attracting audio embeddings and their time-aligned visual context features while repelling the audio embeddings from different frames. Due to this similarity, the novelty of the proposed method may be diminished, resulting in limited contributions. We suggest the authors revise their paper according to the review feedback.